# Blinded Oral Challenges with Lactose and Placebo Accurately Diagnose Lactose Intolerance: A Real-Life Study

**DOI:** 10.3390/nu13051653

**Published:** 2021-05-13

**Authors:** Alba Rocco, Debora Compare, Costantino Sgamato, Alberto Martino, Luca De Simone, Pietro Coccoli, Maria Laura Melone, Gerardo Nardone

**Affiliations:** Department of Clinical Medicine and Surgery, Gastroenterology, University Federico II of Naples, 80131 Naples, Italy; a.rocco@unina.it (A.R.); debora.compare@unina.it (D.C.); costantino.sgamato@unina.it (C.S.); alberto-martino@libero.it (A.M.); luca.desimone1989@gmail.com (L.D.S.); pietro.coccoli@unina.it (P.C.); marialaura.melone33@gmail.com (M.L.M.)

**Keywords:** lactose malabsorption, lactose intolerance, placebo challenge, hydrogen breath test

## Abstract

Lactose intolerance (LI) is characterized by diarrhea, abdominal pain, or bloating occurring after lactose consumption in patients with lactose malabsorption. The National Institute of Health (NIH) proposed a double-blind placebo testing to identify LI individuals correctly. However, until now, no study used this approach in a real-life setting. We aimed to assess double-blind placebo challenge accuracy in diagnosing LI in patients with self-reported symptoms of LI. 148 patients with self-reported LI were consecutively enrolled and blindly underwent hydrogen breath test (HBT) after 25 g lactose or 1 g glucose (placebo) load. One week later, the subjects were challenged with the alternative substrate. Each subject completed a validated questionnaire, including five symptoms (diarrhea, abdominal pain, vomiting, bowel sounds, and bloating) scored on a 10-cm visual analog scale. Home questionnaire (HQ) referred to symptoms associated with the consumption of dairy products at home, while lactose questionnaire (LQ) and placebo questionnaire (PQ) referred to symptoms perceived throughout the 4-h after the administration of the substrates, respectively. After lactose load, HBT was positive in 81 patients (55%), of whom 60 (74%) reported relevant symptoms at LQ (lactose malabsorbers, LM). After placebo challenge, 45 out of 60 with a positive lactose challenge did not complain of symptoms and therefore were diagnosed as lactose intolerant, according to NIH definition. The blinded oral challenges with lactose and placebo accurately diagnose LI and identify patients who will likely benefit from a lactose-free diet.

## 1. Introduction

Lactose intolerance (LI) is a clinical syndrome characterized by diarrhea, abdominal pain, or bloating occurring after lactose consumption in patients with underlying lactose malabsorption. The onset of gastrointestinal symptoms mainly depends on the fermentation of undigested lactose generating volatile gases (hydrogen, carbon dioxide, methane) and short-chain fatty acids by the colonic gut microbiota. Although the genetically-programmed decrease in the lactase enzyme expression occurs in up to 70% of adults worldwide, many individuals with lactose malabsorption do not complain of symptoms after ingesting a standard serving of dairy products [1,2]. Indeed, several additional factors such as type of meal, amount of lactose consumed, the composition of the intestinal microbiome, or previous abdominal surgery can influence the likelihood of developing symptoms [3]. Psychosocial stress and functional gastrointestinal disorders can also impact the subjective perception of LI [4,5].

Currently, the diagnosis of LI needs evidence of lactose malabsorption, assessed by the hydrogen breath test (HBT), associated with the presence of symptoms scored by a validated questionnaire [6,7]. Nevertheless, a non-organic component responsible for the onset of the symptoms due to negative expectations cannot be ruled out, even in patients with a positive lactose tolerance test leading to an overdiagnosis of LI and unnecessary elimination of dairy products from the diet [8].

A consensus conference from the National Institute of Health (NIH) proposed to define LI as the onset of gastrointestinal symptoms following a blinded, single-dose challenge of ingested lactose not observed after the ingestion of an indistinguishable placebo by an individual with lactose malabsorption [9]. 

Blinded testing could represent the ideal method for unmasking the correlation between self-reported symptoms of LI and the objective findings of lactose malabsorption in clinical practice. However, until now, only one study analyzed the prevalence of LI in a young, healthy cohort of subjects using this approach [10]. 

Based on these premises, we designed this prospective study to assess blinded placebo-controlled testing accuracy in diagnosing LI in patients with self-reported symptoms in a real-life clinical setting.

## 2. Materials and Methods

### 2.1. Patients

Consecutive patients referred to our Gastroenterology Unit between January 2018 and June 2019 with an indication to perform lactose HBT because of symptoms consistent with LI were eligible for the study. 

Exclusion criteria were age <18 or >75 years, previous diagnosis of gastrointestinal disease (coeliac disease, Crohn’s disease, or ulcerative colitis), history of abdominal surgery, concomitant severe systemic illness, diabetes mellitus, thyroid disease, pregnancy or breast-feeding and use of prokinetics, antibiotics, prebiotics, probiotics, laxatives, and proton pump inhibitors in the previous four weeks.

### 2.2. Study Protocol

At the time of the study’s inclusion, demographic and anthropometric data and medical history were collected for all patients. Body mass index was calculated as weight in kilograms divided by height in meters squared. On the same day, each patient blindly performed HBT after consuming a solution containing 25 g lactose or 1 g glucose (placebo). One week later, patients repeated the same protocol with the alternative substrate for the HBT. 

Each patient completed a validated questionnaire about symptoms related to habitual consumption of milk-based products at home (home questionnaire, HQ) and throughout the 4-h test after lactose (lactose questionnaire, LQ) or placebo administration (placebo questionnaire, PQ) [7]. One of the investigators (CS), blinded to the experimental substrate used to perform HBT, administered all questionnaires to minimize possible bias sources.

### 2.3. Hydrogen Breath Test

Patients were asked to have 24 h carbohydrate-restricted, low fiber diet and fast in the last 12 h before the test to diminish basal hydrogen excretion. Before starting the test, patients washed their mouths with 20 mL of 0.05% chlorhexidine. Smoking and physical exercise were not allowed for 30 min before and during the test. End-alveolar breath samples were collected immediately before the ingestion of an oral load of 25 g lactose or 1 g glucose (placebo) dissolved in 200 mL water, and breath samples were collected every 30 min for 4 h. 

A numeric sequence randomized the subjects to receive either lactose (solution 1) or glucose as a placebo (solution 2) in the first challenge. The solutions were prepared every day and given to patients in non-transparent glasses with plastic cover by a nurse blinded to the content. One week later, the subjects were challenged with the alternative solution. Both solutions present similar organoleptic characteristics, particularly sweetness.

The H_2_ and CH_4_ concentration in breath samples was measured in parts per million (ppm) using a Quintron Model Breath Tracker DP Microlyzer gas chromatograph (Quintron Instruments, Milwaukee, WI, USA). Lactose malabsorption was diagnosed when H_2_ increased ≥20 ppm or CH_4_ ≥10 ppm over baseline values for a single time point [11]. A baseline H_2_ value ≥10 ppm was defined as an exclusion criterion.

### 2.4. Symptom’s Questionnaire

A validated self-administered questionnaire for LI was used for symptom assessment [7]. The questionnaire includes five items related to symptoms most frequently reported by LI patients (diarrhea, abdominal pain, vomiting, audible bowel sounds, and bloating). Symptom severity was self-rated by the subjects on a 10-cm visual analog scale (VAS) ranging from 0 (without symptoms) to 10 (maximum severity symptoms). The global symptom score (GSS) was the sum of the five VAS individual results (score range 0–50). The LQ and PQ were completed throughout the 4-h test. The symptoms were considered clinically relevant when the GSS was >7.

### 2.5. Statistical Analysis

According to their distribution, assessed with Kolmogorov-Smirnov test, continuous variables were expressed as mean ± standard deviation or median and interquartile range (IQR). Categorical variables were expressed as frequencies. Student t-test was used to compare data expressed as mean ± SD and Mann–Whitney test to compare data expressed as median and interquartile range. Fisher’s exact test was used to compare data expressed as a percentage. Spearman correlation coefficient was calculated to test the association between the overall symptom score and the delta increase in breath H_2_ or CH_4_ levels after the lactose load.

A *p*-value < 0.05 was set as the level of significance. All statistical procedures were performed using SPSS version 20.0 for Windows (SPSS Inc., Chicago, IL, USA). GraphPad Prism version 5.00 for Windows (GraphPad Software, La Jolla CA, USA) was used to create the artworks.

## 3. Results

(1)Overall, female sex was prevalent (100 out of 148 patients, 68%).(2)HBT was positive (Lactose Malabsorbers, LM) in 55% (*n* = 81, of whom 15 methane producers) and negative (Lactose absorbers, LA) in 45% (*n* = 67) of subjects after lactose load. Figure 1 shows the results of 25 g lactose HBT expressed as medians of H_2_ and CH_4_ excretion in ppm at each time point.

Detailed data of the two groups of subjects are shown in Table 1.

(3)All patients reported clinically relevant symptoms (GSS >7) at the HQ. GSS, diarrhea, and abdominal pain mean scores were significantly higher in LA than in the LM group (*p* = 0.005, *p* = 0.001, and *p* = 0.006, respectively) (Figure 2A–C).(4)At the LQ, a significantly higher percentage of LM patients reported a GSS >7 in comparison to LA (74% vs. 42%, *p* = 0.001), although the mean symptom score did not significantly differ between the groups (13.9 ± 9.6 in LM versus 12 ± 9.6 in LA, *p* = 0.244) (Figure 2A). When we analyzed single symptoms, LM complained of vomiting and bloating more frequently than LA (22% vs. 9%, *p* = 0.042 and 81% vs. 64%, *p* = 0.024, respectively). Moreover, bloating scored significantly higher in LM than in LA (5, IQR: 1–8 in LM versus 3, IQR: 0–6 in LA; *p* = 0.021) (Figure 2B–F).(5)At the PQ, 15 out of 81 (18%) LM and 24 out of 67 (36%) LA (*p* = 0.024) had a GSS > 7. The frequency distribution of patients with diarrhea, abdominal pain, vomiting, bowel sounds, and bloating did not significantly differ between the groups. However, abdominal pain severity scored significantly higher in LA than in LM (0, IQR: 0–2 in LM vs 2, IQR: 0–5 in LA; *p* = 0.007) (Figure 2C).

(6)The distribution of GSS at LQ, according to lactose HBT results expressed as delta increase of H2 or CH4 levels in the breath, is shown in Figure 3. Interestingly, there was no significant correlation between GSS and H2 or CH4 delta increase in the breath (*r* = 0.204; *p* = 0.06).

(7)After the placebo challenge, 45 out of 60 (75%) LM patients with GSS > 7 at LQ, did not complain of clinically relevant symptoms, thus fulfilling the criteria for the diagnosis of lactose intolerance, according to NIH Consensus on Lactose Intolerance and Health (Figure 4).

## 4. Discussion

In clinical practice, we often deal with patients who describe a stereotyped symptoms pattern, including abdominal pain, bloating, and diarrhea, after ingestion, even in small amounts, of milk or milk-derivatives products. Although lactose maldigestion symptoms and objective findings are poorly correlated, 20% of the overall population self-report LI, likely reinforced by uncountable articles in media news, social media items, or advertisements for lactose-digestive aids [12]. Consequently, patients tend to unnecessarily eliminate milk and dairy products from their diet with adverse health and psychological implications [13,14].

Several studies well demonstrated that patients misperceive LI [4,15,16]. The pivotal study by Suarez et al. found that 30 subjects who self-reported a severe LI could tolerate up to 240 mL of milk daily (equivalent 12.5 g of lactose), irrespective of lactose digestion phenotype [15]. Later, by using a validated questionnaire, Casella et al. observed that symptoms attributed to LI in everyday life scored significantly higher than those experienced after 50 g lactose challenge, regardless of HBT result [4].

In our study, the number of patients who self-reported clinically relevant symptoms at HQ was significantly higher than that who complained of symptoms after the lactose challenge (100% vs. 70%; *p* < 0.0001). Moreover, all symptom questionnaire items, except for vomiting, scored significantly higher in HQ than LQ, irrespective of lactose HBT result, thus confirming previous reports. The mechanisms that contribute to amplify the perception of symptoms in the home environment are not fully understood. Since we excluded the presence of confounding overlooked conditions, such as celiac disease or inflammatory bowel disease, it is conceivable that additional nutrients such as fats or carbohydrates other than lactose in the meal, [17] or the different composition of the milk compared to lactose with water can, at least in part, explain these findings.

As expected, after the lactose load, LM patients reported clinically relevant symptoms in a percentage significantly higher than LA, although the mean symptom score did not significantly differ between the groups. This result contrasts with previous studies reporting more severe symptoms in LM than LA after lactose ingestion [4,10]. In 353 patients referred for a lactose HBT, after a 50-g lactose load, the total symptom score was significantly higher in 164 patients with true lactose malabsorption than in patients without lactose malabsorption [4]. In a cohort of 121 Chilean youths volunteers, genotyped for the lactase C > T-13910 variant and tested for lactose malabsorption using the HBT with 25 g lactose load, the median symptom score was significantly higher in LM than in LA [9] (IQR 3.25–15) versus 1 (IQR 0–5), respectively [10]. The different amount of lactose used as HBT substrate and the type of populations enrolled in our study likely account for the discrepancies in the results. Twenty-one out of 81 (26%) LM patients did not experience clinically relevant symptoms after lactose ingestion. They likely had a higher tolerance threshold, and 25 g lactose used in our study was insufficient to induce symptoms [18].

Using the placebo challenge, only 45 out of 60 patients who fulfilled the diagnostic criteria of LI, according to LQ and HBT results, were symptom-free. Thus, according to the definition proposed by the expert panel of NIH Consensus on Lactose Intolerance and Health, LI prevalence in our population of patients with self-reported LI was 30%. On the other hand, four out of 28 LA patients with GSS > 7 at LQ did not complain of symptoms during the placebo challenge. One possible explanation could be a false-negative breath test result after lactose challenge. The idiopathic absence of H_2_-producing flora occurs in about 10–20% of normal subjects, usually producing other gases such as methane and hydrogen sulfide. The combined H_2_ and CH_4_ excretion measurements we used in our study, enhancing HBT accuracy, reduce the risk of underdiagnosing lactose malabsorption [11,19,20]. However, since we did not measure hydrogen sulfide in the breath, we cannot exclude that these patients were hydrogen sulfide-producers.

Interestingly, 15 LM and 24 LA with a GSS > 7 at LQ complained of clinically relevant symptoms even after the placebo challenge. A non-organic component, such as negative expectations, conditioning, anxiety, and somatization, predominant in LA patients, could explain these results [21,22]. Thus far, a “nocebo effect”, i.e., the occurrence of symptoms following the administration of an inert substance, along with the suggestion that the subject will get worse, has been described in a considerable proportion of patients undergoing lactose HBT [8,23]. On the other hand, the belief that lactose-containing foods may lead to abdominal symptoms, powerfully influenced by mass-media, is so frequent in the general population to prompt a growing expansion of the lactose-free dairy market, expected to reach a nine billion turnover by 2022 [24]. Unfortunately, we do not have information about the usage of lactose-free products in our population. Literature data about the food choices of LI people is scarce. In a survey conducted in the US, 75% of LI people avoided dairy, with over half of them worrying about the long-term risks to their health due to this dietary restriction [24]. Thus, it seems reasonable to recommend lactose restriction only to patients fulfilling NIH criteria for LI, who likely will have a positive impact on symptoms.

In our population, the female sex was significantly prevalent. However, evidence to support any unique associations between LI symptoms and gender is insufficient. Indeed, the amount of lactose ingested, dietary pattern, and lactase non-persistence are the primary drivers of the intolerance symptoms [25]. Thus, it is likely that gender-specific behavior and perception of risk account for the higher percentage of women in our study.

We are aware that the lack of lactase-gene polymorphisms evaluation could represent a limitation in our study. However, most studies found an excellent agreement between genetics for lactase and HBT in diagnosing lactase deficiency [26,27,28]. Furthermore, the use of a symptom questionnaire validated for 50 g lactose dose could have reduced its sensitivity in identifying LI patients in our study. On the other hand, the percentage of LM patients with a positive LQ (77%) reported by Casellas et al. [7] does not differ from that observed in our study (74%). Finally, a symptom questionnaire validated for the 25 g lactose load, which is the recommended dose by international guidelines to diagnose lactose malabsorption, is still lacking and remains an urgent need in this research area.

Noteworthy, to the best of our knowledge, this is the first study that analyses the blinded-testing approach to diagnoses LI in a real-life clinical setting. Furthermore, the stringent inclusion criteria, the rigorous double-blind placebo approach, the combined H_2_ and CH_4_ measurements in HBT, strengthen our results.

## 5. Conclusions

Our study demonstrated that the blinded oral challenge with lactose and placebo is a feasible and useful approach to diagnosing LI. Should our data be confirmed in larger, multicenter studies, the use of a questionnaire validated for 25 g lactose load and the double-blinded testing at one-week intervals could be recommended in patients with self-reported LI in the routinary clinical practice.

## Figures and Tables

**Figure 1 nutrients-13-01653-f001:**
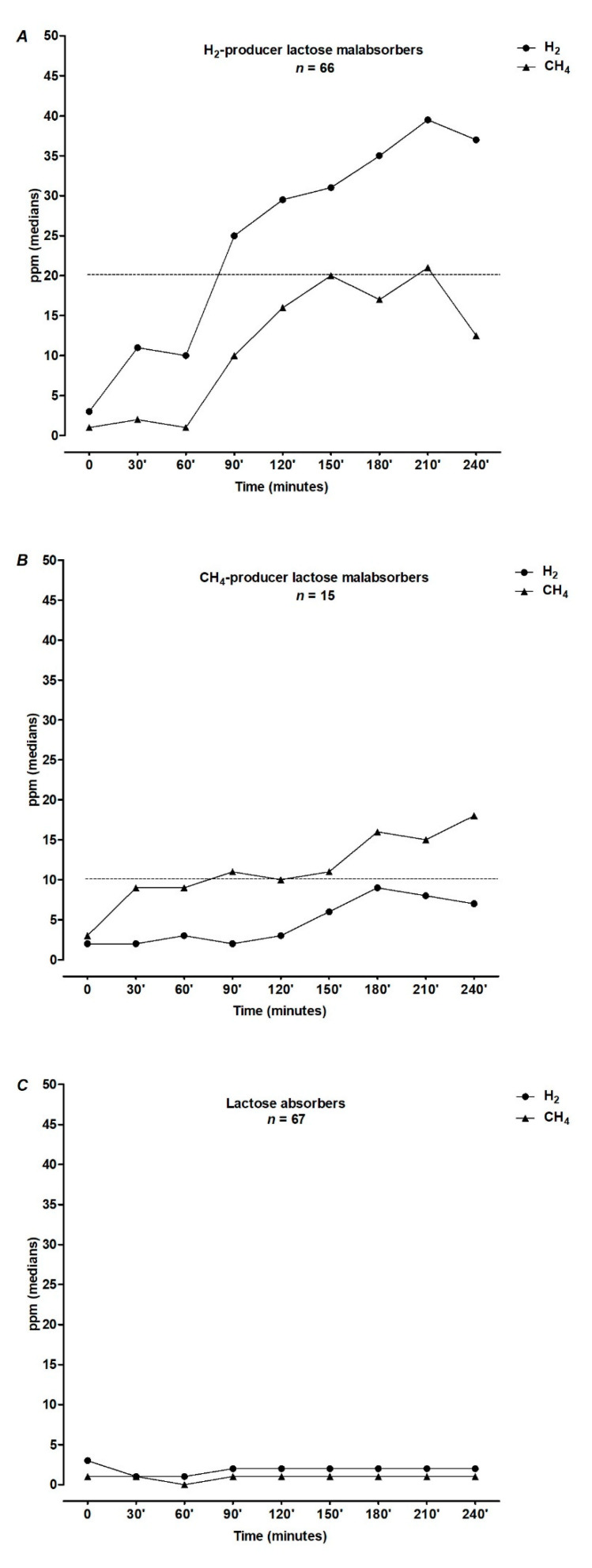
Results of 25 g lactose hydrogen breath test expressed as medians of H_2_ and CH_4_ excretion in ppm at each time point. The dotted lines indicate, respectively, the currently effective cut-off values of H_2_ or CH_4_ to diagnose lactose malabsorption.

**Figure 2 nutrients-13-01653-f002:**
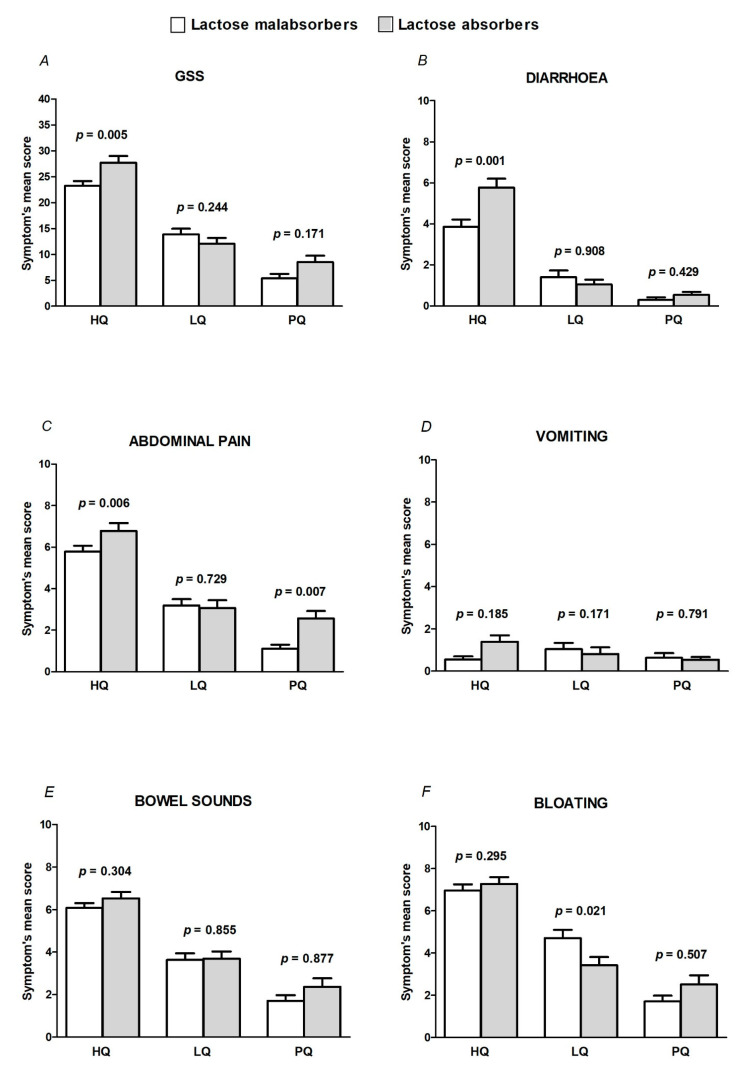
Global symptom score (GSS) (**A**) and diarrhea (**B**), abdominal pain (**C**), vomiting (**D**), bowel sounds (**E**), and bloating (**F**) mean scores in the studied population, subdivided according to lactose hydrogen breath test results. Lactose malabsorbers (HBT positive after lactose challenge), lactose absorbers (HBT negative after lactose challenge), respectively. HQ: home questionnaire; LQ: lactose questionnaire; PQ: placebo questionnaire.

**Figure 3 nutrients-13-01653-f003:**
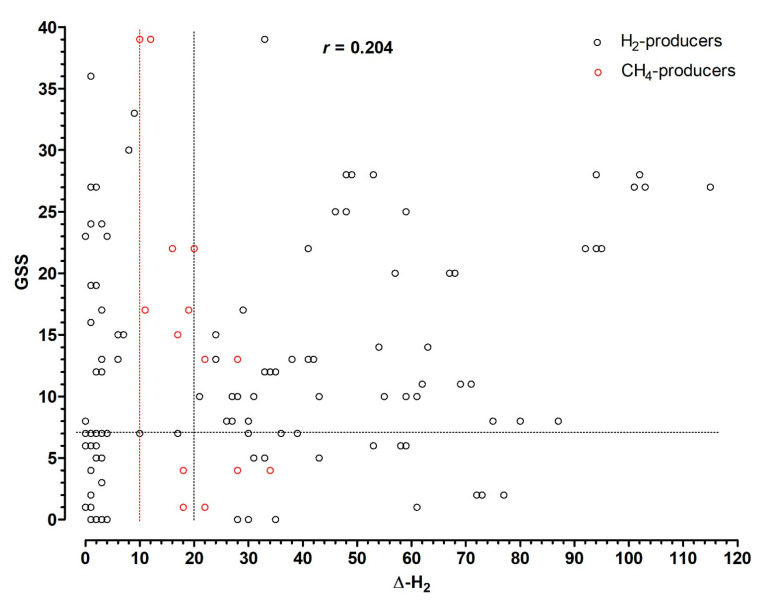
Global symptom score (GSS) distribution according to lactose hydrogen breah test (HBT) results, expressed as delta increase over baseline of H_2_ or CH_4_ levels in the breath after 25 g lactose load. The horizontal dotted line indicates the cut-off of clinically relevant GSS. The vertical dotted red and black lines indicate the cut-offs used to diagnose lactose malabsorption for CH_4_ and H_2_, respectively. There was no significant correlation between GSS and H_2_ or CH_4_ delta increase in the breath (*r* = 0.204; *p* = 0.06).

**Figure 4 nutrients-13-01653-f004:**
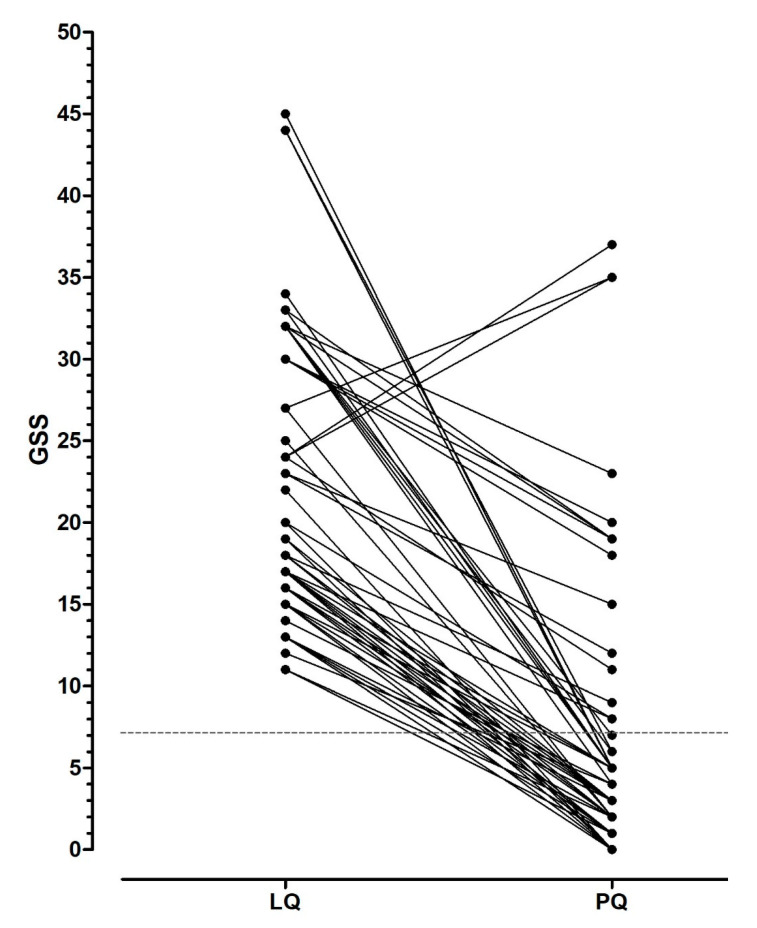
Global symptom score (GSS) after 25 g lactose load and placebo in the subgroup of patients classified as lactose malabsorbers according to lactose hydrogen breath test and abnormal symptom score after lactose load. Note that, after the placebo challenge, 15 patients still complain of clinically relevant symptoms. The dotted line indicates the cut-off value used to define the clinically relevant score of symptoms. LQ: lactose questionnaire, PQ: placebo questionnaire.

**Table 1 nutrients-13-01653-t001:** Demographic, anthropometric, and clinical findings of the population enrolled in the study subdivided according to lactose hydrogen breath test results.

Patients	LM (*n* = 81)	LA (*n* = 67)	*p* *
Age	41.8 ± 17.3	40 ± 18	0.7252
Males/Females	18/63	30/37	0.0081
BMI, ^1^ Kg/m^2^	24 ± 3.8	24.2 ± 5.2	0.8709
^§^ Global Symptom Score >7			
Home questionnaire	81 (100%)	67 (100%)	1.000
Lactose questionnaire	60 (74%)	28 (42%)	0.001
Placebo questionnaire	15 (19%)	24 (36%)	0.024

^1^ BMI: body mass index. * t-test and Fisher’s exact test were used to compare data expressed as mean ± SD and data expressed as percentages, respectively. ^§^ Global Symptom Score is the sum of five symptom scores (diarrhea, abdominal pain, vomiting, audible bowel sounds, and bloating). A Global Symptom Score > 7 identifies the patients with clinically relevant symptoms.

## Data Availability

The data that support the findings of this study are available from the corresponding author upon reasonable request.

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
