# Peer review of "Blinded Oral Challenges with Lactose and Placebo Accurately Diagnose Lactose Intolerance: A Real-Life Study"

_nutrients, 2021, doi:10.3390/nu13051653_

Round 1
Reviewer 1 Report
It is a work that deals with a topic of great interest in which the authors insist on the need to avoid the systematic exclusion of milk and milk derivatives in patients who report symptoms and suspect that they are lactose intolerant. The study design is better than that of previous studies and the authors are aware of its limitations and so report it in the discussion and conclusions. However, the results are not clearly stated and the interpretation of some tables and figures is difficult. In my opinion, a major drawback is the administration of only 25 grams of lactose and the use of a validated questionnaire for the administration of 50g of it. The results are inconclusive and they only allow to report that, indeed some patients with a positive result in the intestinal gas production after the administration of lactose do not report symptoms and while others also report them after the administration of the placebo.
Reviewer 2 Report
The paper described lactose-intolerance diagnosis via placebo-controlled HBT in human patients with self-assessed LI. The content is clinically relevant, the results interesting and important to the field.
Major criticism:
- Please describe methods clearly, also in Abstract (Time points of 2 HBT, questionnaire assessment when)
- State clearly how the results should be transferred into clinical practice, e.g. use standardized questionnaires AND do HBT after placebo-challenge AND do lactose-challenge; at which time intervals?
- Please mention in discussion whether usage of lactose-free products a) was performed by these patients, b) was effective or not in these patients, c) should be/or not be recommended to which patient group?
Minors:
- line 7: delete duplicate of "correspondence"
- line 13: consecutive = consecutive appearance at clinic with presumed LI?
- line 14: "items" is "symptoms"?
- line 16: put comma after "(HQ)," otherwise sentence is unclear
- line 19-20: clarify that placebo-challenge was negative in 45/60 but lactose-challenge was positive in the same patients, and therefore they were diagnosed as LI.
- line 52: add "placebo-controlled" testing
- line 69: "consuming" instead of assuming
- line 69: state gain the amount of lactose as well as the substrate and amount of placebo
- line 78: delete "lactose" in subtitle, as placebo was also done
- Figure 1: clarify what each data point represent (median of all tested patients; n=.....)
- line 128: dotted lines indicate the "currently effective" cut-off values of H2...
- Table 1: please clarify of which total number the percent numbers are representative, e.g. heading: total (n= 148); LM (n=...); LA (n=...)
- Table 1, questionnaires:
- are numbers in brackets again percentages?
- what is presented here by "home questionnaire", "LQ" and "PQ" - symptom appearance? of how many symptoms?
- placebo-questionnaire shows sign. p-value - what does that imply? disuss.
- males and females obviously show differences (p-value sign) - discuss
- line 135: should LA and LM be changed?
- line 138: change "in respect" to "in comparison"
- Figure 2: for better clarity, please insert legend for white/grey columns also in diagrams; clarify in line 147 again, whether LM and LA here means that HBT after lactose-challenge was positive or negative, respectively
- line 190: instead of "in respect" write "compared"
- lines 210-213: unclear: 4/28 did not complain during placebo-challenge - should they have?
- line 232: better "between genetics for lactase and HBT"
Round 2
Reviewer 1 Report
I think the article can be accepted now